# Titanium(IV) Oxo-Complex with Acetylsalicylic Acid Ligand and Its Polymer Composites: Synthesis, Structure, Spectroscopic Characterization, and Photocatalytic Activity

**DOI:** 10.3390/ma15134408

**Published:** 2022-06-22

**Authors:** Julia Śmigiel, Tadeusz Muzioł, Piotr Piszczek, Aleksandra Radtke

**Affiliations:** Faculty of Chemistry, Nicolaus Copernicus University in Toruń, Gagarina 7, 87-100 Toruń, Poland; juliasmigiel@doktorant.umk.pl (J.Ś.); tadeuszmuziol@wp.pl (T.M.)

**Keywords:** titanium(IV) oxo-complex, acetylsalicylic acid, molecular structure, spectroscopic characterization, photocatalytic activity

## Abstract

The titanium oxo complexes are widely studied, due to their potential applications in photocatalytic processes, environmental protection, and also in the biomedical field. The presented results concern the oxo complex synthesized in the reaction of titanium(IV) isobutoxide and acetylsalicylic acid (Hasp), in a 4:1 molar ratio. The structure of isolated crystals was solved using the single-crystal X-ray diffraction method. The analysis of these data proves that [Ti_4_O_2_(O^i^Bu)_10_(asp)_2_]·H_2_O (**1**) complex is formed. Moreover, the molecular structure of (**1**) was characterized using vibrational spectroscopic techniques (IR and Raman), ^13^C NMR, and UV–Vis diffuse reflectance spectroscopy (UV–Vis DRS). The photocatalytic activity of the synthesized complex was determined with the use of composite foils produced by the dispersion of (**1**) micrograins, as the inorganic blocks, in a polycaprolactone (PCL) matrix (PCL + (**1**)). The introduction of (**1**) micrograins to the PCL matrix caused the absorption maximum shift up to 425–450 nm. The studied PCL + (**1**) composite samples reveal good activity toward photodecolorization of methylene blue after visible light irradiation.

## 1. Introduction

Photocatalytic technologies are an important direction of current research, with many practical applications. The issue of photocatalysis, used to remove contaminants from air and water, self-cleaning, or surface disinfection, caused more interest in recent years [1,2]. Materials based on titanium dioxide belong to the intensely investigated systems, with regard to their use in catalysis, photochemistry, environmental protection, and biomedicine [2,3,4,5,6,7,8,9]. Titanium(IV)-oxo (TOCs) clusters also belong to this group of materials, and their structural diversity and unique physicochemical and optical properties are of particular interest for their potential applications in photocatalytic technologies [3,10,11,12]. The advantage of TOCs is the possibility of physicochemical property modifications associated with changes of the {Ti_a_O_b_} core structure and the functionalization way of stabilizing ligands (L) [13,14]. The synthesis of titanium(IV)-oxo clusters, of the general formula [Ti_a_O_b_(OR)_x_L_y_] (a ≥ 2, b = 2a − x/2 − y/2, x ≥ 1, y ≥ 1), is based on the reaction of titanium alkoxides (Ti(OR)_4_; R = Et, ^i^Pr, ^i^Bu, ^t^Bu) with carboxylic acids, phosphinates, phosphonates, or Schiff bases (L). Among the various synthesis strategies of TOCs, the most important is the traditional synthesis in strict inert conditions (Schlenk line, RT, glove box, inert atmosphere) (a), and the solvothermal strategy (b) [15,16,17]. The revision of the reaction conditions, e.g., the Ti:L stoichiometric ratio, solvents, and reaction temperature, allow for the control of the [Ti_a_O_b_(OR)_x_L_y_] cluster structure [11]. The important issue of works carried out are studies on oxo clusters consisting of {Ti_a_O_b_} cores containing a = 3 or 4 Ti atoms. These types of clusters are crucial in the synthesis of TOCs because the base units in the next stages of their synthesis can lead to the {Ti_a_O_b_} core enlargement (a > 4) [17,18,19,20,21]. On the other hand, studies on their electronic structure and photocatalytical applications are also exciting. The previous works considering [Ti_4_O_2_(O^i^Bu)_10_(O_2_R′)_2_] clusters (R′ = -*m*-PhCl, -*m*-PhNO_2_, *-p*-PhNH_2_, and -C_13_H_9_) prove that carboxylate ligands functionalization allows the modulation of the HOMO–LUMO gap value in the range 3.59–2.55 eV [14,19]. Simultaneously, the UV–Vis DRS measurements prove the shift of the absorption maximum from the UV range towards the visible one, which is connected with a shift of TOCs photocatalytic activity shift into this range [8]. Continuing the previous works on the photocatalytic activity of tetranuclear Ti(IV)-oxo clusters, research on their synthesis using acetylsalicylic acid (Hasp) was carried out. The received results are presented herein. 

Acetylsalicylic acid (aspirin—asp) is a commonly used agent, with antipyretic, anti-inflammatory, and anticoagulant effects [22,23,24]. The biological activity of aspirin is due to the actions of the molecule’s acetyl and salicylate portions [24]. Therefore, it was interesting to investigate the possibility of using this compound to synthesize Ti(IV)-oxo complexes. The presented paper shows the results concerning the synthesis, structural characterization, and electronic structure of oxo clusters stabilized by asp ligands. Moreover, the issues associated with photocatalytic activity of the composite films produced by the dispersion of synthesized oxo clusters in poly(ε-caprolactone) matrix are discussed. A poly(ε-caprolactone) (PCL) is a biodegradable thermoplastic polymer increasingly used in the production of medical devices [25,26]. Since the PCL matrix may undergo degradation in water surroundings, it was important to study the stability of the composite sample during photocatalysis processes. Our investigations aim to develop composite coatings, and exhibit their photocatalytic activity in the visible light range. 

## 2. Materials and Methods

### 2.1. Materials

Titanium(IV) isobutoxide (Aldrich, St. Louis, MO, USA) and acetylsalicylic acid (ASA, Aldrich, St. Louis, MO, USA) were purchased commercially, and were used without further purification. All solvents used in the synthesis, i.e., tetrahydrofuran (THF) and isobutanol (HO^i^Bu) were distilled before their use, and stored in argon atmosphere. The processes of Ti(IV)-oxo complexes synthesis were carried out using the standard Schlenk technique, in the inert gas atmosphere (Ar) and at room temperature (RT).

### 2.2. Synthesis of the Tetranuclear Ti(IV)-Oxo Complex (1) Stabilized by Acetylsalicylic Ligands and the PCL + (1) Composite

The synthesis of the tetranuclear Ti(IV)-oxo complex stabilized by acetylsalicylic ligands (**1**): 0.16 g of acetylsalicylic acid (0.875 mmol) was added to the solution of 1.19 mL titanium(IV) isobutoxide (3.5 mmol), in 2 mL of THF/HO^i^Bu (1:1), leading to a clear yellow solution. The solution was left for crystallization. The light yellow crystals of [Ti_4_O_2_(asp)_2_(Bu^i^O)_10_]·H_2_O were isolated from the mother liquor after three days; yield: 74%. Elemental analyses were performed on Elemental Analyser vario Macro CHN (Elementar Analysensysteme GmbH, Langenselbold, Germany). Titanium content was gravimetrically determined as TiO_2_, according to the Meth-Cohn et al., method [27]. The results of elemental analysis: calculated, C = 52.4%, H = 8.0%, Ti = 14.5%; experimental, C = 52.7%, H = 7.8%, Ti = 14.9%.

The composite films containing 10, 15, and 20 wt.% of isolated Ti(IV)-oxo complex micrograins were prepared, by an addition of (**1**) (0.10, 0.15, and 0.20 g) dispersed in 1 mL of THF to the polycaprolactone (PCL) solution (1.0 g of PCL dissolved in 5 cm^3^ of THF). The resulting mixtures were stirred in an ultrasonic bath for 120 min. In the next step, they were poured into a glass Petri dish, and left for the evaporation of the solvent at RT in the inert atmosphere (glove box). The composite films of 50 µm thickness were characterized by Raman and IR spectroscopy, and scanning electron microscopy.

### 2.3. Analytical Procedures

#### 2.3.1. Single Crystal X-ray Diffraction Measurement

The diffraction data of (**1**) were collected at 100 K on MX14-2 beamline of BESSY II (Helmholtz Zentrum, Berlin, Germany) synchrotron. The data were processed using xdsapp software [28,29], and, subsequently, CrysAlis Pro (Applied Rigaku Technologies, Inc., Austin, TX, USA) [30] was used to apply the numerical absorption correction. The structure was solved by the direct methods and refined with full-matrix least-squares procedure on F^2^ (SHELX-97 [31]). Heavy atoms were refined with anisotropic displacement parameters, whereas hydrogen atoms were assigned at calculated positions with thermal displacement parameters fixed to a value of 20% or 50% higher than those of the corresponding carbon atoms. In the final model there were missing hydrogen atoms from water molecules. We also observed positional disorder for five isobutanolate anions. For stability of the refinement process, several restraints, mainly on thermal parameters (ISOR), were applied. Additionally, EADP constraints were used for disordered O^i^Bu- anions. All figures were prepared in DIAMOND (CRYSTAL IMPACT, Brandenburg GbR, Bonn, Germany) [32], ORTEP-3 (Department of Chemistry, University Of Glasgow, Glasgow, UK) [33], and CrystalExplorer [34,35,36]. The results of the data collections and refinement are summarized in Table 1, selected bond lengths and angles are presented in Appendix A. CCDC 2158421 contains the supplementary crystallographic data for (**1**). These data can be obtained free of charge from the Cambridge Crystallographic Data Centre, via www.ccdc.cam.ac.uk.data_request/cif.

#### 2.3.2. Spectroscopic Characterization

The structures of the isolated solid reaction products (crystals and powders) were confirmed using vibrational spectroscopy methods, i.e., IR spectrophotometry (Perkin Elmer Spectrum 2000 FTIR spectrophotometer (400–4000 cm^−1^ range, KBr pellets)) and Raman spectroscopy (RamanMicro 200 spectrometer (PerkinElmer, Waltham, MA, USA)). Raman spectra were registered using a laser with the wavelength 785 nm, with a maximum power of 350 mW, in the range 200–3200 cm^−1^, using a 20 × 0.40/FN22 objective lens, and an exposure time of 15 s each time. The ^13^C NMR spectra in the solid phase were recorded on a Bruker Advance 700 (Madison, WI, USA) 700 MHz spectrometer, with a spectral width of 76,923.08 Hz and 4096 complex points. Thermogravimetric analysis coupled to an infrared spectrophotometer was performed using an instrument Bruker Optik (Ettingen, Germany). Measurements were carried out in the range 20–1300 °C, with a heating speed of 5 °C/min in the nitrogen atmosphere.

The HOMO–LUMO gap energy values of isolated complex was determined by using diffuse reflectance UV–Vis spectra (UV–Vis DRS), which were registered between 200 and 800 nm. A Jasco V-750 spectrophotometer was used (JASCO Deutschland GmbH, Pfungstadt, Germany). The recorded spectra were evaluated in terms of energy band gap values, via Spectra Manager TM CFR software.

The produced PCL + (**1**) foil surfaces were studied using a scanning electron microscope with field emission (SEM, Quanta 3D FEG, Houston, TX, USA). Composite materials underwent thermal treatment (Bruker Optik, Ettingen, Germany) in the range 20–500 °C, with the heating speed of 5 °C/min in the nitrogen atmosphere. Based on the UV–Vis DRS spectra registered between 200 and 600 nm, the absorption maximum of the produced PCL + (**1**) composite films was determined.

#### 2.3.3. The Photocatalytic Activity Evaluation of (PCL + TOCs) Composites

The photocatalytic activity of PCL+ (**1**) foil was studied by monitoring the degradation processes of methylene blue solution (MB), according to ISO 10678:2010 procedure [37,38]. Foil samples of sizes 1 × 1 cm were preconditioned by exposure to visible light for 30 h. In the next step, foils were placed in plastic cuvettes with MB solution (V = 3.5 cm^3^ and C = 2.0 × 10^−5^ M). After 12 h in the dark, the solutions were replaced by the appropriate test of MB solutions (V = 3.5 cm^3^ and C = 1.0 × 10^−5^ M). The prepared samples were exposed to visible light in the range 395–425 nm (integrated LED 395–425 nm, 700 mA, 30 W). MB absorbance at 660 nm was registered (Metertech SP-830 PLUS, Metertech, Inc., Taipei, Taiwan) after 0 and 0.5–70 h (measuring every 1 h) of irradiation. Percentage of MB decolorization was calculated using the equation: % dye decolorization = ((C_0_ − C_t_)/C_0_) × 100 = ((A_0_ − A_t_)/A_0_) × 100)(1)
where C_0_ is an initial concentration of dye, C_t_ is a dye concentration at a given time *t*, and A_0_ and A_t_ are absorbances at 0 and t times, respectively [37,39].

## 3. Results

### 3.1. Structure of [Ti_4_O_2_(asp)_2_(Bu^i^O)_10_]·H_2_O (asp = O_2_C-o-PhO_2_CCH_3_) (1)

The molecular structure of (**1**) was solved using the single-crystal X-ray diffraction method, and the received results are presented in Figure 1 and Appendix A. The selected bond lengths and angles found in the structure of (**1**) are listed in Appendix A. The analysis of collected data reveals that the structure description of the (**1**) cluster could use the following general formula; [Ti_4_O_2_(asp)_2_(Bu^i^O)_10_]·H_2_O. The {T_i4_O_2_} cores are stabilized by two acetylsalicylate and four isobutoxide bridges, and also six terminal isobutoxide groups (Figure 1a). The postulated positions of oxygen atoms attributed to water molecules are presented in Figure 1b. The analysis of a Hirshfeld surface suggests the formation of the weak C85-H85E…O9[−1/2+x, 1/2−y, −1/2+z] hydrogen bond (Figure 2). 

### 3.2. NMR Spectroscopy

Carbon-13 nuclear magnetic resonance spectroscopy (^13^C NMR) was also used to structure the isolated crystals (**1**) characterization. Considering the results of the chemical shift, values of coordinated asp and -O^i^Bu ligands are attributed as follows: (a) Ti_2_-O_2_C-Ph-O-C-CH_3_ group: Ph at 165.9, 133.2, 119.0 ppm, O_2_C- at 177.1 ppm, Ph-O-C(O)- at 187.3 ppm, and -CH_3_ at 25.0; (b) Ti-O-CH_2_-CH-(CH_3_)_2_ group: -CH_2_- at 86,83 ppm, -CH- at 30.7 ppm, and -CH_3_ at 19.11 ppm; (c) THF (solvent) at 25.5, 69.3 ppm (Appendix A). 

### 3.3. Analysis of Vibrational Spectra

The structure of (**1**) was also characterized using IR and Raman spectra, and the received results are presented in Table 2, Appendix A. The water molecules present in the structure of this compound are evident by the broad band of the low intensity, attributed to ν(OH) stretching modes, which registers in the range 3100–3400 cm^−1^ of the IR spectrum. Moreover, the weak band at 3080 cm^−1^ in the Raman spectrum is also attributed to the ν(OH) stretching modes of water molecules [40]. The bands between 1400 and 1800 cm^−1^ (registered in IR and Raman spectra) are assigned to stretching vibrations of coordinated carboxylate ligands (ν_as_(COO) and ν_s_(COO)), and carbonyl (ν(C=O)) of the ester group (Table 2). Moreover, bands derived from the stretching vibrations of ν(C=C) phenyl groups are detected in this range of the IR spectrum [41]. The strong bands around 1096 and 1026 cm^−1^ are assigned to ν(CO) stretching vibrations of the coordinated terminal and bridged alkoxide ligands [20]. Considering the results of our earlier DFT calculation of the normal vibrations of titanium oxo bridges in [Ti_4_O_4_(OR)_10_(O_2_CR′)_2_] clusters, the bands registered between 519 and 708 cm^−1^ in IR and Raman spectra of (**1**) are attributed to stretching vibrations of ν(Ti-(μ-O)-Ti) and ν(Ti-(μ_4_-O)-Ti) bridges (Table 2) [14]. 

### 3.4. UV–Vis Diffuse Reflectance Spectra (UV–Vis DRS) of the Oxo-Complex and HOMO–LUMO Gap Determination

UV–Vis DRS spectra of the oxo-complexes were registered at room temperature, using magnesium oxide as a standard reference (Figure 3). The HOMO–LUMO gap values were determined based on the Kubelka–Munk (K–M) function versus light energy, i.e., K = f(hν), where K = (1 − R)^2/^2R and R is the reflectance, which was used for the optical band gap determination (Figure 3b) [42,43]. According to the obtained data, the tested oxo-complex shows absorption at the limits of UV and Vis in the range of λ_max_ = 395 nm. The value of the energy gap of the HOMO–LUMO is 2.35 eV.

### 3.5. The Films of Poly(Caprolactone) and (1) Composite

The produced microcrystalline powder of (**1**) was dispersed in a polycaprolactone (PCL) matrix, thus, forming composite films (PCL + (**1**)), which contain 10, 15, and 20 wt.% of this compound. Analysis of SEM images exhibit the relatively homogeneous distribution of TOCs grains in the PCL matrix (Figure 4). The dispersion of (**1**) grains in PCL matrix was also investigated using Raman microscopy (Figure 5). The use of this method allows for the determination of the dispersion of oxo-cluster grains in the PCL matrix, and confirms the structural stability of the complex (**1**) after its introduction into the above-mentioned matrix.

### 3.6. Thermal Analysis of PCL + (1) Composites

Possible changes in thermal properties of the studied composites, caused by the addition of TOCs, were estimated using thermogravimetric analysis (TGA) and differential scanning calorimetry (DSC). The measurements were carried out in the temperature range 30–600 °C, in nitrogen atmosphere, and the obtained results are presented in Table 3 and Appendix A. The analysis of TGA data of all tested samples enriched with the oxo complex shows a weight loss in the temperature range of 384–395 °C. It is a one-step process, where weight loss is between 81 and 89%. Compared to the pure PCL, it is seen that the addition of (**1**) micrograins shifts the decomposition temperature, so we obtain greater thermal stability of the produced composite.

The pure PCL thermogram shows a single endothermic peak at about 58 °C (*T_m_*), attributed to the polymer’s melting. The dispersion of the (1) grains in the PCL matrix slightly influences the melting point. The second endothermic peak, *T_d/max_*~368 °C, is attributed to the degradation of the polymer. In the case of PCL + (1) composites, a clear increase in the decomposition temperature is observed.

### 3.7. Estimation of Photocatalytic Activity of the Oxo-Complexes

UV–Vis DRS spectra of the composite samples were recorded, using magnesium oxide as a standard at room temperature. Data presented in Figure 6 show changes in the absorption maximum position for PCL samples enriched with 10, 15, and 20 wt.% (**1**), compared to the unenriched PCL matrix. According to these data, we note that the addition of the oxo-complex shifts the absorption maximum towards visible light. In all examined cases, this point falls between 425 and 450 nm (Figure 6). The photocatalytic activity was estimated based on methylene blue solution (MB) photodecolorization, during irradiation with visible light by 70 h. The results of these investigations prove that all composite samples reveal photocatalytic activity. Almost 100% of MB solution decolorization is noticed for the PCL + (**1**) 15 wt.% sample, while for PCL + (**1**) 10 and 20 wt.% samples it is c.a. 89 and 95 wt.%, respectively (Table 4). For comparison, the photodecolorization of MB and PCL measured under the same conditions amounts to c.a. 33 and 37%. Due to the possible instability of the hydrolytic of (**1**), and the degradation processes of the PCL film in aqueous solutions, we performed the photodecolorization of the MB solution twice for the same sample of the PCL + (**1**) 15 wt.% (Figure 7b). The analysis of received data reveals that no significant loss of activity is observed for the tested composite sample. Figure 8 shows the IR spectra of the PCL + (**1**) 15 wt.% sample recorded before and after 140 h of the MB solution photodegradation experiment. The comparison of IR spectra registered composite samples spectra of PCL and (**1**) indicate the structural stability of (**1**), and the beginning of PCL degradation after 140 h of the sample’s interaction with the aqueous MB solution. Moreover, spectral changes in the composite spectrum after 140 h of the photocatalysis process, characteristic of the degradation of the PCL matrix, are also found.

## 4. Discussion

The crystals of titanium(IV)-oxo cluster, with the general formula [Ti_4_O_2_(O^i^Bu)_10_(O_2_CR′)_2_]·H_2_O (asp = O_2_C-o-PhO_2_CCH_3_) (**1**), were isolated. The mother liquor was a 4:1 mixture of titanium(IV) isobutoxide and acetylsalicylic acid, in the 1:1 THF/HO^i^Bu mixture as solvent (room temperature, inert atmosphere). The results of the conducted research confirm that the use of a 4:1 molar ratio (titanium alkoxides and organic acid), and the above-mentioned reaction conditions, leads to the formation of oxo cluster, which consists of {Ti_4_O_2_} cores [14,19].

The research carried out reveals that the molecular structure of (**1**) consists of {Ti_4_O_2_} cores stabilized by two carboxylate (asp), four isobutoxide bridges, and six terminal isobutoxide groups (Figure 1). Analysis of bond lengths and angles presented in Appendix A reveals that the isolated oxo complex forms a similar type of tetranuclear cluster as found in previously described systems [Ti_4_O_2_(O^i^Bu)_10_(O_2_CR′)_2_] (R′ = C_13_H_9_, PhCl, PhNO_2_, PhNH_2_) [14,19,21]. The appearance of water molecules in the crystal lattice of (**1**) is an interesting issue from the structural point of view. Generally, the synthesis of this compound is carried out under anhydrous conditions. According to the mechanism postulated for synthesizing Ti(IV)-oxo complexes, the source of water molecules is the esterification process [17]. The presence of water molecules in the crystal lattice of (**1**) may indicate that during the next processes, i.e., hydrolysis and condensation, not all of the H_2_O molecules formed undergo these reactions. The weak bands found at 3100, 3280 cm^−1^, and 3080 cm^−1^ in the IR and Raman spectra of (**1**), respectively, confirm the presence of water molecules in this structure (Table 2). Analysis of structural data indicates the dense packing of tetranuclear Ti(IV)-oxo clusters in the crystal lattice of (**1**), which form relatively weak mutual interactions (Appendix A). There are localized aggregates in the empty spaces between four clusters, consisting of four water molecule positions (Figure 1b and Appendix A). The analysis of their mutual interactions cannot be completed, due to missing hydrogen atoms from water molecules.

Nevertheless, a Hirshfeld surface analysis, with a blue colour denoting distances longer than the van der Waals radii sum, suggests the formation of hydrogen bonds (Figure 2a). The red spots indicating short contacts are related to the shortest H…H contacts between terminal BuiO- anions. On the fingerprint presented in Figure 2b, they occur as spikes at ca. (1.0, 0.9; 1.1, 0.9; 0.9, 1.0; 0.9, 1.1). The received data suggests that O…H contacts are significantly longer, and they should be attributed to a rather weak C85-H85E…O9 [-1/2+x, ½-y, -1/2+z] hydrogen bond between the methyl group of the disordered O81 isobutanolate, and O9 atom from the esther group of O1 carboxylate (Figure 2c). This is the only hydrogen bond detected in the reported structure. Hence, the oxygen atoms of ester groups, localized in the ortho position in the aspirin ligands (as a potential proton acceptor), are buried deeply in the cluster, and are found in the unfavorable positions. C…H interactions cover a small portion of contacts. However, they seem important, as C3 and C13 phenyl rings are involved in two C-H…π and two π-π interactions. During our earlier investigations, we draw attention to the possibility of water molecules binding (formed during ethyl esterification) in the crystal lattice of titanium(IV)-oxo clusters [43]. However, in the previous case, H_2_O molecules were trapped in a “cage”, formed by three titanium atoms and three terminals -O^i^Bu ligands. 

Analyzing the NMR data, attention is drawn to the presence of additional peaks, attributed to the THF solvent (Appendix A). Generally, it should be noted that structural and spectroscopic (IR, Raman) data do not indicate the presence of THF molecules in the structure (**1**). Although, in the synthesis of this compound, the 1:1 THF/HO^i^Bu mixture was used as a solvent. Therefore, we believe this effect is associated with an incomplete solvent removal from the studied sample. Confirmations are the thermogravimetric analysis results of (**1**), and the IR spectra of the volatile thermolysis products of this compound in the range of 35–150 °C (Appendix A). According to these data, the thermal decomposition of (**1**) starts right after heating, and is related to the detachment of H_2_O molecules and alcohol species (transition 35–150 °C). This is confirmed by the IR spectra of volatile thermolysis products of (**1**) (Appendix A), in which bands assigned to water appear in the ranges of 3400–3800 cm^−1^ and 1200–1900 cm^−1^. Bands that appear at ~3450 cm^−1^, ~2990, 2800 cm^−1^, and 1030 cm^−1^ are attributed to stretching vibrations of the -OH, -CH, and -C-O groups, respectively, and suggest the detachment of alcohol species. If the THF appears in the gas phase in the range of 35–150 °C, the bands at ~2970, 2860 cm^−1^, ~1100, and 900 cm^−1^ assigned to the -CH, C-O-C, and C-C stretching vibrations, respectively, should appear.

The UV–Vis DRS measurements prove the position of the absorption maximum (λ_max_) at 395 nm, i.e., on the border between UV and visible range. According to previous reports, the functionalization of stabilizing ligands in oxo clusters structures is important for shifting the absorption maximum from the UV range towards visible one. A similar effect is also observed by Chaumont et. Al. for the group of [Ti_10_O_12_(cat)_8_®_8_] clusters, where R denotes pyridine and substituted pyridines [44]. The influence of the functionalization way of the carboxylate ligand, stabilizing the {Ti_a_O_b_} core, on the position of λ_max,_ and the value of the energy gap (*E*_a_), is also noted by Liu et al. [45,46]. According to their results, the introduction of various carboxylate ligands to labile coordination sites of the hexanuclear [Ti_6_O_4_(O^i^Pr)_10_(O_3_P-Phen)_2_(OAc)_2_] complex allows the modulation of the bandgap values in 3.6–3.0 eV range. Studies on [Ti_4_O_2_(O^i^Bu)_10_(O_2_R′)_2_] clusters (R′ = -m-PhCl, -m-PhNO_2_,-p-PhNH_2_, and -C_13_H_9_) reveal the changes of the band gap value in the similar range, i.e., 3.59–2.55 eV [14,19]. In the case of (**1**), which forms a similar molecular structure type, the energy bandgap is 2.35 eV (Figure 3), and is lower than in above-mentioned tetranuclear oxo clusters. This energy bandgap decrease is important for the photocatalytic activity of this compound, especially for its shifts towards the visible range. Therefore, in all our photocatalytic experiments, samples were irradiated by visible light between 395 and 425 nm. Interestingly, the manufacture of photoactive materials in the visible range is important for both photocatalytic and antimicrobial applications [47,48]. 

Due to the hydrophobic nature of (**1**), and its possible sensitivity to hydrolysis processes in all our photocatalytic experiments, we used composite films (PCL + (**1**)) produced by the dispersion of (**1**) micrograins as an inorganic component in poly(ε-caprolactone) (PCL) matrix. 

Analysis of SEM images and Raman maps support the uniform distribution of (**1**) grains in the composite films (Figure 4 and Figure 5). The Raman spectra of the PCL + (**1**) composite samples, registered at different surface points, confirms that after the dispersion of (**1**) in the polymer matrix, its molecular structure does not change in a significant way (Figure 5). The use of the UV–Vis DRS method reveals that the addition of (**1**) to the PCL matrix shifts the absorption maximum from the UV range (c.a. 240 nm for the pure PCL) up to the visible one (c.a. 450 nm for the PCL + (**1**) composite). Thermal analysis (DSC and TGA) of the produced composites containing 10, 15, and 20 wt.% of (**1**) prove that the introduction of the inorganic component to the polymer matrix increases the decomposition temperature of the composite by 27–33 °C (DSC) in comparison to pure PCL (Table 3). 

The use of multinuclear Ti(IV)-oxo complexes in photodegradation processes of organic dyes is an issue that is intensively studied in recent years [49,50,51]. Considering the measured energy gap value for (**1**) microspheres (2.35 eV), and the position of maximum absorption for the PCL + (**1**) composite at 420 nm, the photoinduced activity of composite samples was determined by examining the MB solution photodecolorization process when irradiated with violet/blue light. In our experiments, we used PCL + (**1**) composites containing 10, 15, and 20 wt.% of (**1**). The studied composite films reveal good photocatalytic activity toward MB; however, their activity slightly changes with the content of TOCs in the polymer matrix. Table 4 shows the MB decolorization percent on levels 89, 95, and 100% for 10, 20, and 15 wt.% content of (**1**) in the PCL matrix, respectively. A double repetition of the photocatalytic experiments in the same conditions for PCL + (**1**) 15 wt.% sample reveals no significant activity loss, which may indicate the structural stability of (**1**) towards photocatalytic processes. The results of research on the photocatalytic activity of the PCL + (**1**) composite may be compared with the results of our previous investigations concerning PCL + TOCs composites (TOCs = [Ti_4_O_2_(O^i^Bu)_10_(O_2_CR′)_2_] (R′ = PhNH_2_ and C_13_H_9_)) [8]. However, in our previous tests, the composite fittings (produced by the injection molding method) were used, and the photocatalytic process was conducted for 30 h, in visible light. The comparison of the obtained results shows that after 30 h, the photocatalytic activity of PCL + (**1**) is comparable to the PCl + ([Ti_4_O_2_(O^i^Bu)_10_(O_2_CNH_2_)_2_]) system, and weaker than for PCl + ([Ti_4_O_2_(O^i^Bu)_10_(O_2_CC_13_H_9_)_2_]). However, if we compare the test results of PCL + (**1**) film samples with PMMA + TOCs films (TOCs = [Ti_4_O_2_(O^i^Bu)_10_(O_2_CR’)_2_] (R′ = PhNH_2_ and C_13_H_9_), PMMA = poly(methyl methacrylate)), no significant differences are found.

Registering the IR spectra of PCL + (**1**) composite before and after 140 h of the photodecolorization process of the MB aqueous solution, clear differences are noted. Analysis of IR spectra, presented in Figure 8, indicate that the structure of (**1**) during the whole photocatalytic experiment does not change. Proof of this is the presence of weak and medium intense bands at c.a. 1550, 1590 cm^−1^ (coordinated carboxylate groups), between 1000 and 1150 cm^−1^ (coordinated alkoxide groups and COC vibrations of the ester groups), and in the 510–760 cm^−1^ range (vibrations of Ti-(μ-O)-Ti and Ti-(μ_4_-O)-Ti bridges, and C-C vibrations of the ortho substituted benzene ring). Simultaneously, the lack of a clear band between 550 and 800 cm^−1^ in the IR spectrum of the composite sample after the photocatalytic experiment confirms the lack of the possible TOCs hydrolysis product, i.e., titanium dioxide. For comparison, the IR spectra of PCL + TiO_2_ 0.5 wt.% system, complex (**1**), PCL, and the composite PCL + (**1**) 15 wt.% are presented in Appendix A. Two bands appearing in the PCL + TiO_2_ composite spectrum at c.a. 720 and 560 cm^−1^ are assigned to the stretching vibrations of titanium dioxide [52,53]. These bands are superimposed on the PCL polymer bands, their degradation products, and the complex (1) appearing in this range. Therefore, we assume that the hydrolysis of (**1**) results in the appearance of a clear band at 720 cm^−1^, which is not found in our spectra of PCL + (**1**) composites after the photocatalytical process (140 h) (Figure 8). On the other hand, in the IR spectrum of the PCL + (**1**) composite after photocatalytic experiments (140 h), the effects associated with the beginning of PCL degradations appear. In general, during the hydrolytic degradation of PCL, we observe the breaking of polymer chains according to the following scheme [54,55]:~[~O-(CH_2_)_5_-C(O)-O-(CH_2_)_5_-C(O)~]~ + H_2_O → ~[~O-(CH_2_)_5_-C(O)-OH + HO-(CH_2_)_5_-C(O)~]~

The consequence of this process is the appearance of carboxylate and hydroxy groups at the ends of the polymer chains. The IR spectrum of the PCL + (1) composite, recorded after 140 h of the photocatalytic process, becomes more complicated (Figure 8). The bands assigned to O-H stretching vibrations were found in the range 3000–3700 cm^−1^. A significant broadening of the 1774/1762 cm^−1^ band of the carbonyl group of ester chains is caused by the appearance of the band c.a. 1700 cm^−1^, which is assigned to the asymmetric stretching vibrations of the -COOH group. The complexity of this spectrum also results from: (a) the effects related to the appearance of bands assigned to symmetrical vibrations of the -COOH groups (below 1450 cm^−1^); (b) the -OH groups bending modes (between 1300 and 1400 cm^−1^); and (c) the disappearance of bands originating from COC vibrations of the ester groups (between 1000 and 1150 cm^−1^). Similar changes are found in IR spectra of the PCL, registered before and after photocatalytic processes (Appendix A). A clear difference is noted in the range 3100–3700 cm^−1^, in which bands of stretching vibrations of H_2_O molecules appear (Figure 8 and Appendix A). Their intensity is significantly greater in the composite spectrum than in the PCL one. The noticed effect may be the result of the difference in the way both samples are dried, or the nature of the composite film degradation in MB solution. The process flow of PCL + TOCs composite degradation will be investigated in detail in the next stage of our work.

## 5. Conclusions

During the investigations carried out, a [Ti_4_O_2_(O^i^Bu)_10_(asp)_2_] · H_2_O (**1**) cluster was synthesized, the structure of which was solved by single-crystal X-ray diffraction. The oxo complex (**1**) was also characterized by vibrational spectroscopy (IR and Raman), and solid-state ^13^C NMR spectroscopy. Analysis of the UV–Vis DRS spectrum proves the maximum absorption position at c.a. 395 nm, and the value of the HOMO–LUMO energy band gap on the level 2.35 eV. In studies on photocatalytic activity, the possible sensitivity of (1) to hydrolysis processes causes the composite samples produced by dispersion of the oxo complex in the poly(ε-caprolactone) (PCL) matrix (PCL + (1)), which are then used. The introduction of (**1**) microcrystals to the polymer matrix causes: (a) an increased thermal stability of PCL + (1) composite in comparison to pure PCL, and (b) a significant shift of the absorption maximum from the UV range up to the visible range. Simultaneously, the use of Raman spectroscopy confirms the structural stability of (**1**) during the formation of PCL + (**1**) composite and degradation of the PCL matrix, after 140 h of the photocatalytic experiment. The photocatalytic experiments reveal good PCL + (**1**) film photocatalytic activity in the visible range. This activity does not change during the degradation process of the PCL matrix, which may result from the structural stability of the oxo cluster (**1**). 

## Figures and Tables

**Figure 1 materials-15-04408-f001:**
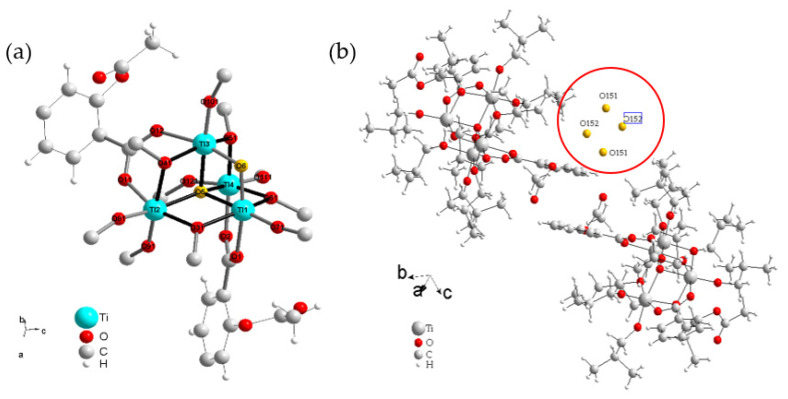
(**a**) Ti_4_O_2_ core of [Ti_4_O_2_(asp)_2_(Bu^i^O)_10_]·H_2_O (**1**) with labeled titanium and oxygen atoms. For clarity of the picture, all hydrogen atoms and the most of carbon atoms are omitted. Oxygen atoms from oxo bridges are given in yellow, from carboxylate anions in orange, from Bu^i^O^−^ anions in red. (**b**) Positions of water molecules oxygen atoms (yellow and circled) presented in the crystal lattice of (**1**).

**Figure 2 materials-15-04408-f002:**
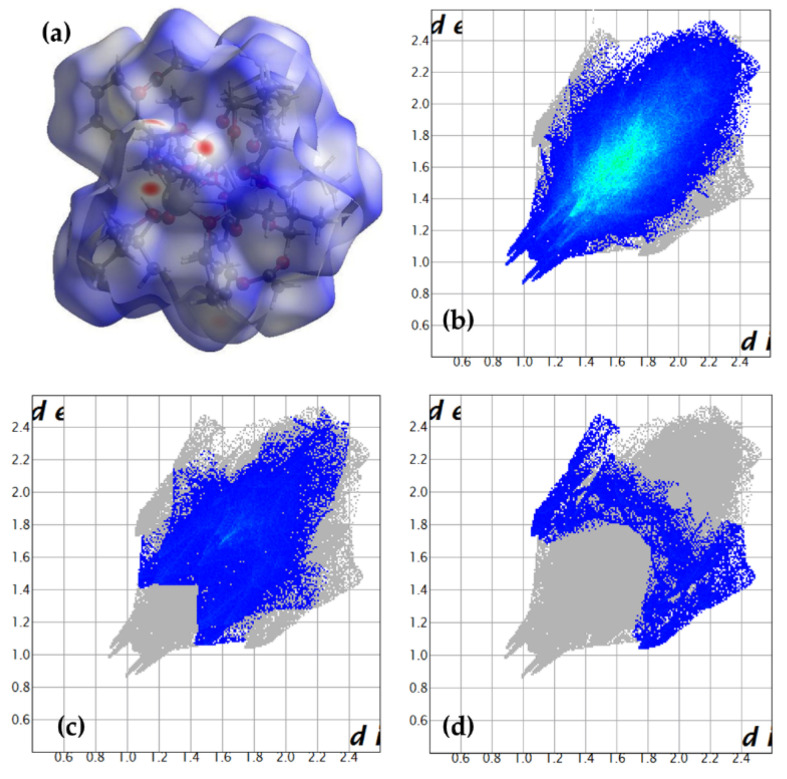
(**a**) Hirshfeld surface with projected all interactions shows prevailing blue color. The fingerprints for (**b**) H…H (83.5%), (**c**) O…H (11.7%), and (**d**) C…H (3.9%) are also given.

**Figure 3 materials-15-04408-f003:**
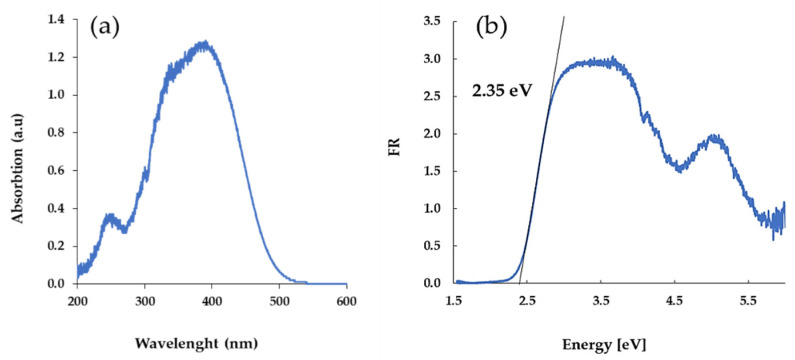
The solid-state UV–Vis diffuse reflectance spectra (DRS) of the micrograins (**a**), and light energy plot for the HOMO–LUMO gap determination (**b**).

**Figure 4 materials-15-04408-f004:**
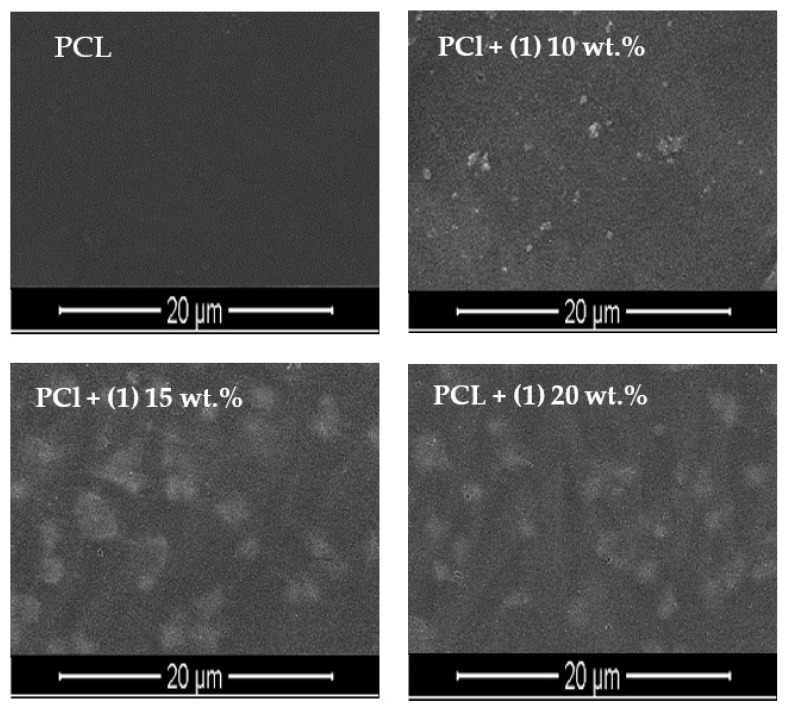
SEM images of PCl and PCL + (**1**) *n* wt.% composites (*n* = 10, 15, and 20 wt.%).

**Figure 5 materials-15-04408-f005:**
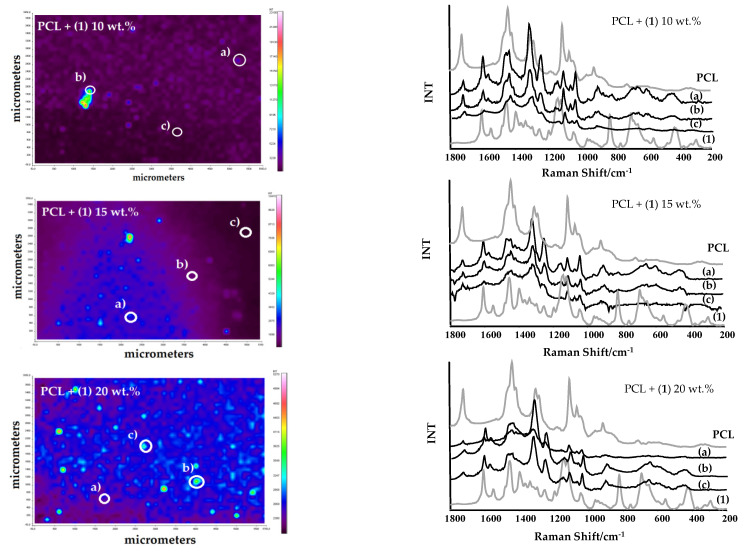
Raman microscopy maps and Raman spectra registered in selected points of the PCL + (**1**) composites.

**Figure 6 materials-15-04408-f006:**
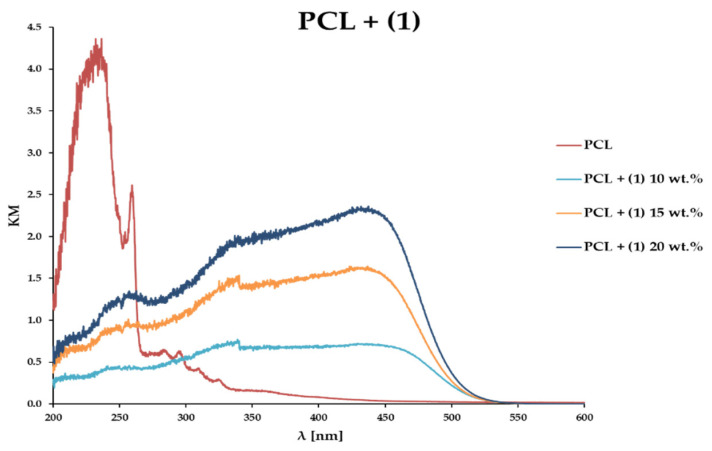
UV–Vis DRS spectra of PCL and studied PCL + (**1**) composites.

**Figure 7 materials-15-04408-f007:**
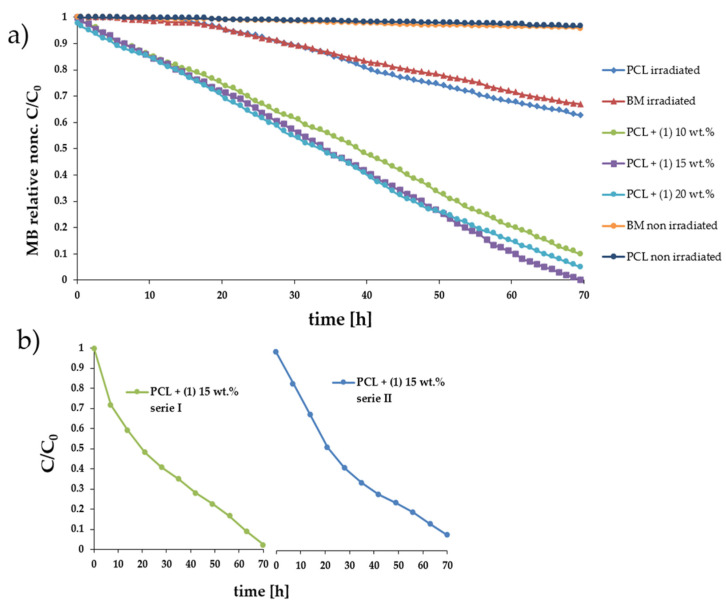
(**a**) Photodecolorization of the methylene blue (MB) solutions as a function of time for the respective PCL + (**1**) composite films containing 10, 15, and 20 wt.% of (**1**), and irradiated with visible light. (**b**) The recyclability study of PCL+ (**1**) 15 wt.%.

**Figure 8 materials-15-04408-f008:**
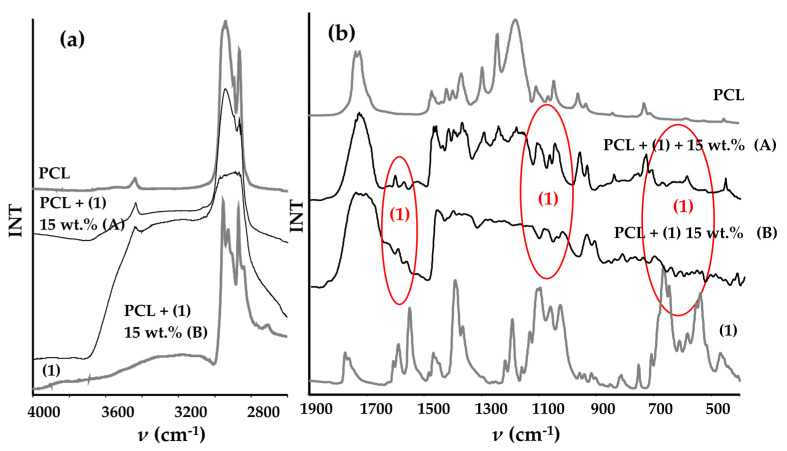
IR spectra registered in the ranges 2600-4000 cm^−1^ (**a**) and 400-1900 cm^−1^ (**b**) of PCL + (**1**) 15 wt.% composite samples before (**a**), and after (**b**) the photocatalytical experiment, during which they were immersed in the MB solution for 140 h. Composite spectra are compared with the PCL and (**1**) IR spectra (the locations of the bands characteristic for the (**1**) micrograins are marked in red).

**Table 1 materials-15-04408-t001:** Crystal data and structure refinement for (**1**).

Empirical Formula	C_58_ H_106_ O_21_ Ti_4_
Formula weight	1331.02
Temperature	100(2) K
Wavelength	0.85506 Å
Crystal system	Monoclinic
Space group	P2_1_/n
Unit cell dimensions [Å] and [°]	a = 13.092(3) α = 90°. b = 24.028(5) β = 97.09(3)°. c = 22.752(5) γ = 90°
Volume [Å^3^]	7102(3)
Z, calculated density [Mg/m^3^]	4, 1.245
Absorption coefficient [mm^−1^]	0.777
F(000)	2840
Crystal size [mm^3^]	0.120 × 0.080 × 0.060
Theta range for data collection	1.489 to 32.304°.
Index ranges	−16 ≤ h ≤ 16 −30 ≤ k ≤ 30 −28 ≤ l ≤ 28
Reflections collected/unique	88738/14372 [R(int) = 0.0320]
Completeness to theta = 30.866°	99.0%
Absorption correction	Numerical
Max. and min. transmission	1.000 and 0.231
Refinement method	Full-matrix least-squares on F^2^
Data/restraints/parameters	14372/68/833
Goodness-of-fit on F^2^	1.051
Final R indices [I > 2 sigma (I)]	R1 ^a^ = 0.0687, wR2 ^b^ = 0.2051
R indices (all data)	R1 ^a^ = 0.0704, wR2 ^b^ = 0.2068
Largest diff. peak and hole	1.272 and −0.865 e. Å^−3^

^a^ R1 = Σ‖*F*_0_| − |*F_c_*‖/Σ|*F*_0_|; ^b^ wR2 = [Σw(*F*_0_^2^ − *F_c_*^2^)^2^/Σ(w(*F*_0_^2^)^2^)]^1/2^.

**Table 2 materials-15-04408-t002:** Positions of the selected bands registered in infrared (IR) and Raman spectra of (**1**) (very weak (vw), weak (w), medium (m), strong (s)).

Modes	(1)	
	IR (cm^−1^)	R (cm^−1^)
ν(OH) (water molecules)	3280 (w)	-
	3100 (w)	-
ν(OH) (water molecules)	-	3080 (w)
ν(CH)	3080 (w)	-
ν(C=O) (ester group)	1774 (m)	-
	1762 (m)	1759 (vw)
ν(C=C)	1605 (w)	1608 (s)
ν_as_(COO) (carboxylate group)	1589 (m)	-
	1550 (s)	1551 (m)
ν_s_(COO) + ν(CC) (carboxylate group)	1483 (w)	1484 (w)
	1467 (m)	-
	1459 (w)	1459 (m)
	1447 (w)	-
	-	1400 (s)
ν(C-O) (ester group)	1216 (m)	1225 (w)
ν(C-O) (alkoxide group)	1096 (s)	-
ν(COC) (ester group)	1050 (m)	-
ν(C-O) (alkoxide group)	1026 (s)	-
ν(CC)	753 (m)	755 (w)
ν_a_(Ti-(μ-O)-Ti)	708 (m)	-
		692 (s)
δ(CH)	683 (m)	676 (m)
	666 (s)	

ν_a_(Ti-(μ_4_-O)-Ti) +	650 (s)	649 (m)
δ(OCO) (ester)	611 (w)	600 (w)

	583 (m)	
ν_a_(Ti-(μ_4_-O)-Ti)	552 (s)	
ν_s_(Ti-(μ_4_-O)-Ti)		536 (w)
	539 (s)	
	519 (w)	
		513 (vw)

**Table 3 materials-15-04408-t003:** Thermal parameters received from thermogravimetric analysis (TGA) and differential scanning calorimetry (DSC) of the composites (*T_m_* = melting temperature, *T_d_* = decomposition onset temperature, *T_d/max_* = decomposition temperature, ∆m = thermal transition weight loss).

	DSC		TGA	
Composite			Stage	Solid Residue
	*T_m_*/°C	*T_d_*/°C	*T_dmax_*/°C	at 450 °C (%)
PCL	41.2	318.2	347	2.11
PCL + (**1**) 10 wt.%	40.3	344.6	384	10.81
PCL + (**1**) 15 wt.%	41.8	339.8	395	15.77
PCL + (**1**) 20 wt.%	42	340.7	391	18.03

**Table 4 materials-15-04408-t004:** MB solution decolorization percentages.

Composite	MB Decolorization (%)
MB-irradiated	33.03
PCL	37.1
PCL + (**1**) 10 wt.%	89.22
PCL + (**1**) 15 wt.%	100
PCL + (**1**) 20 wt.%	94.87

Methylene blue (MB) decolorization at the end of the measurements (*t* = 70 h).

## Data Availability

Data is contained within the article or Appendix A.

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
