# Peer review of "Titanium(IV) Oxo-Complex with Acetylsalicylic Acid Ligand and Its Polymer Composites: Synthesis, Structure, Spectroscopic Characterization, and Photocatalytic Activity"

_materials, 2022, doi:10.3390/ma15134408_

Round 1

Reviewer 1 Report

In current manuscript (materials-1754885), the authors reported the fabrication of titanium oxo complex [Ti4O2(OiBu)10(asp)2]·H2O. The as-fabricated complex is characterized by Infra-red, Raman, 13C NMR, and UV-Vis-DRS techniques. The photocatalytic activity of the as-fabricated complex is evaluated for photodecolorization of methylene blue under visible light irradiation. The optimized sample showed 100% decolorization after visible light irradiation for 70 hours. This work is well organized and has scientific value. Therefore, I would like to consider its publication in materials after minor revision as noted below.

Specific comments:

1. In the introduction part, key scientific issue needs to be addressed. For example, environmental concern. Following articles should be useful.

https://doi.org/10.1002/smtd.202101395

https://doi.org/10.1016/j.jechem.2021.08.023

2. The authors should label Figure S2.

3. The authors should mention the source of light used for decolorization of methylene blue.

4. Could the authors explain why it took 70h for decolorization of the methylene blue dye? There are abundant catalysts which can decolorize high concentration dyes in just few hours. Could this material be used in practical applications?  

5. Please carefully check reference number 17, 24, 27, 28, 29, 38, and 42.

Author Response

Thank you for your in-depth review, based on which we have prepared a new version of our article.

Please find attached replies to detailed comments and comments:

  1. In the introduction part, the key scientific issue needs to be addressed. For example, environmental concern.

Considering the reviewer’s recommendations, we have changed the Introduction, by adding the necessary information, including environmental concerns – using the abovementioned articles.

  1. The authors should label Figure S2.

We do not understand this remark. Figure S2 is cited in the manuscript and Figure S2 caption is provided in the Supplementary Materials section.

  1. The authors should mention the source of light used for the decolorization of methylene blue.

Information on the light source used for the decolorization of methylene blue has been added on page 4.

  1. Could the authors explain why it took 70h for decolorization of the methylene blue dye? There are abundant catalysts which can decolorize high concentration dyes in just few hours. Could this material be used in practical applications?

Analysis of the literature data shows that the photocatalytic activity of oxo-clusters is measured by the direct contact of the sample (powders, crystals) with the dye solution (e.g. MB). In such conditions, the photodecolorization rate is greater. However, the problem is the sensitivity of oxo-Ti(IV) clusters to hydrolysis processes. Therefore, we decided to introduce TOCs into the polymer matrix in our research. The dispersion of TOCs in the matrix and the concentration of 10, 15, and 20 wt.% caused that MB photodecolorization slowed down (70h). However the use of a dispersed compound in a polymer matrix makes it possible to use it in a much wider range of applications, for example as surfaces coatings that self-disinfect under the influence of visible light.

  1. Please carefully check reference number 17, 24, 27, 28, 29, 38, and 42.

Reference numbers 17, 24, 27, 28, 29, 38, and 42 have been checked and changed, moreover, all references have been reviewed.

Reviewer 2 Report

The paper submitted by J. Åšmiegel and coworkers reports on a novel Ti(IV) oxo-complex and its use as a photocatalyst dispersed on a PCL matrix. In my opinion, it is a really good paper which deserves to be published after some minor changes.

·       Introduction, lines 25 – 26: The authors write ‘It
causes TOCs arouse interest in their application in catalysis, photochemistry, material science, environmental protection, and biomedicine [3,510].' It is not clear what is meant by 'It'. One sentence above, it is written that ‘The great interest in titanium(IV)
-oxo clusters (TOCs) observed in recent years is due 24
to their structural diversity and unique physicochemical and optical properties [14]’, so it is not quite clear if the word ‘It’ in the second sentence refers to ‘The great interest’, ‘structural diversity’ or ‘unique physicochemical and optical properties’? Please rewrite to make it clear.

·       Introduction, line 31: define ‘traditional synthesis’.

·       Introduction, lines 33 – 39: the subscripted variables a and b in the formulas, e.g. TiaOb, are written partially in italics and partially in normal fonts; please use either one or the other, but not a mix of both

·       Introduction, line 70: ‘The obtained results will develop systems (…)’: firstly, results per se cannot obtain any systems, and secondly, this is a speculation. Pleas rewrite more carefully.

·       Introduction, line 72: Is ‘aquatic’ here really the right term? Aquatic means: relating to water; living in or near water or taking place in water. Aqueous environment would be more appropriate.

·       Materials and Methods, line 88: when giving the calculated results, the proposed formula from which those results were calculated, should be given.

·       Materials and Methods, line 99: the authors should provide a short comment of why the synchrotron measurements were used and not the standard X-ray technique.

·       Table 1: the R indexes seem to be high; please comment on that

·       Results, lines 182, 183 and 197: the term ‘registered’ is rather unusual; ‘measured’ might be more appropriate.

·       Results, line 203: the unit (eV) is missing after 2.35.

·       Results, line 223 and further on: there should be a blank space between the number and the unit °C, e.g., 600°C should be 600 °C.

·       Table 2: T and m should be written in Italic

·       Results, line 252: filml should read film

·       Discussion, lines 275 – 277 and Conclusion, lines 383 – 385: the sentences are almost the same; there is no reason to repeat it twice. Please rewrite or remove one of those sentences either from Discussion or from Conclusion.

Author Response

Thank you for your in-depth review, based on which we have prepared a new version of our article.

Please find attached replies to detailed comments and comments:

  • Introduction; lines 25 – 26: The authors write ‘It
    causes TOCs arouse interest in their application in catalysis, photochemistry, material science, environmental protection, and biomedicine [3,5–10].' It is not clear what is meant by 'It'. One sentence above, it is written that ‘The great interest in titanium(IV)-oxo clusters (TOCs) observed in recent years is due 24
    to their structural diversity and unique physicochemical and optical properties [1–4]’, so it is not quite clear if the word ‘It’ in the second sentence refers to ‘The great interest’, ‘structural diversity’ or ‘unique physicochemical and optical properties’? Please rewrite to make it clear.

      In the revised version of the manuscript this part of the Introduction has been changed.

  • Introduction; line 31: define ‘traditional synthesis’.

     "Traditional synthesis" means the reaction between the titanium(IV) alkoxide and the organic acid carried out in Schlenk's vessel at RT and inert atmosphere. In the next step of the synthesis, the reaction mixture is placed in a glove box (or a vacuum desiccator) and left for crystallization at RT, or in the refrigerator. More detailed information about the traditional synthesis has been added to the text (page 2)

  • Introduction; lines 33 – 39: the subscripted variables a and b in the formulas, e.g. TiaOb, are written partially in italics and partially in normal fonts; please use either one or the other, but not a mix of both

Corrected in accordance with the reviewer's comment.

  • Introduction; line 70: ‘The obtained results will develop systems (…)’: firstly, results per se cannot obtain any systems, and secondly, this is a speculation. Pleas rewrite more carefully.

      In the revised version of the manuscript this part of the Introduction was changed.

  • Introduction; line 72: Is ‘aquatic’ here really the right term? Aquatic means: relating to water; living in or near water or taking place in water. Aqueous environment would be more appropriate.

Corrected in accordance with the reviewer's comment

  • Materials and Methods; line 88: when giving the calculated results, the proposed formula from which those results were calculated, should be given.

        The elemental composition was calculated for the isolated crystals of Ti44-O)(μ-O)(asp)2(BuiO)10]·H2O; this information has been added to the text.

  • Materials and Methods; line 99: the authors should provide a short comment of why the synchrotron measurements were used and not the standard X-ray technique.

     We performed data collection on the synchrotron because of poor diffraction observed on our home diffractometer. The experiment carried out on MX14-2 beamline allowed for a significant shortening of the experiment time. Moreover, a data set revealed much better quality, which resulted in a better structure of the tested oxo complex.

  • Table 1: the R indexes seem to be high; please comment on that

        The main problem that we encounter when studying the crystal structures of TOCs is their structural instability resulting, inter alia, from their complexity and the mechanism of synthesis. This is of particular importance in the case of synthesis using the "traditional" technique (Schenk line, RT, glove box, inert atmosphere). Therefore, we cannot agree that R indexes are high. They are rather moderate, and this model was possible only due to synchrotron radiation. The model obtained using a diffractometer was much worse in terms of R1 and other validation parameters. The presented final model includes some disordered isobutanolate anions, and several restraints were applied to get the stable refinement and reasonable geometry. It also resulted in slightly elevated R indexes which are acceptable, taking into account the complicity of the structure and its instability as well as another model for such core (structure 2 – [Ti4O2(OiBu)10(O2CPhNO2) – Janek et al., Materials 2018, 11, 1661; doi:10.3390/ma11091661).

  • Results, lines 182, 183 and 197: the term ‘registered’ is rather unusual; ‘measured’ might be more appropriate.

According to us we are able to use the term “registered”; in this part of the manuscript, we present the results of the IR, Raman spectra analysis. Spectra are recorded/registered but not measured.

  • Results, line 203: the unit (eV) is missing after 2.35.

Corrected in accordance with the reviewer's comment

  • Results, line 223 and further on: there should be a blank space between the number and the unit °C, e.g., 600°C should be 600 °C.

Corrected in accordance with the reviewer's comment

  • Table 2: T and m should be written in Italic

Corrected in accordance with the reviewer's comment

  • Results, line 252: filml should read film

Corrected in accordance with the reviewer's comment

  • Discussion, lines 275 – 277 and Conclusion, lines 383 – 385: the sentences are almost the same; there is no reason to repeat it twice. Please rewrite or remove one of those sentences either from Discussion or from Conclusion.

In the revised version of the manuscript, this part of the Conclusions was changed.

Reviewer 3 Report

This paper presents a complete study on the synthesis, detailed characterization, and photoactivity performance of its polymer composite of a Titanium(IV) Oxo-Complex. Overall, this paper is clearly written, great novelty, and the photocatalytic performance and stability of the PCL composite is impressive. The reviewer is wondering if this method can be potentially applied to other polyester commodity polymers such as PET? It would be an interesting study to expand the scope of the complex and the method presented in this study. The reviewer recommends publication of this study in Materials in its present form.

Author Response

Thank you for your in-depth review, based on which we have prepared a new version of our article.

Please find attached replies to detailed comments and comments:

The reviewer is wondering if this method can be potentially applied to other polyester commodity polymers such as PET?

Our research is also carried out using PMMA and PS as polymer matrices. The main problem is the processing temperature at which the composites are formed. This temperature should be lower than 100°C during composite formation (thermal stability of TOCs). Our experiments have shown additionally that these complexes cannot be incorporated into epoxy resins due to possible reactions with the hardener.

It would be an interesting study to expand the scope of the complex and the method presented in this study.

Our group study on oxo-Ti(IV) complexes with various core sizes {TiaOb}, as well as with various functionalization of stabilizing ligands. These investigations are carried out to study the influence of the aforementioned factors on the photocatalytic activity of the synthesized compounds and their biological properties. A separate issue that we are investigating is the use of oxo-complexes as systems allowing for the transport of antimicrobial or anti-aggregating factors.

Reviewer 4 Report

The work of J. Åšmigiel, T. MuzioÅ‚, P. Piszczek and A. Radtke is devoted to the synthesis and detailed characterization of a tetranuclear titanium oxo-complex [Ti4O2(OiBu)10(asp)2]·H2O stabilized by acetylsalicylic ligands, as well as the preparation and study of the photocatalytic activity of the composite material obtained by incorporating this complex into a polymer matrix. This is an interesting study, carried out at a good scientific level, corresponding to the recent work of this group of authors. The results are of interest to the readers of the journal Materials, however, a number of points should be revised before publication.

1) The motivation for the introduction of asp-ligands, given in the Introduction (lines 50-64), and its relation with further studies presented in the work, is unclear. 

In particular, the authors assumed the susceptibility of the oxo-cluster to hydrolysis processes, which should lead to the release of acetylsalicylic acid (Hasp) in the biological fluids. Further, the authors present the polymer matrix as a kind of instrument for controlling the release of Hasp. How does this expected benefit relate to the results of photocatalytic activity of composite material presented in this manuscript? However, the release of acetylsalicylic acid into solution from the composite material has not been studied. Therefore, it is not clear how workable this idea is.

Further, the general idea is not clear. Why is it necessary to introduce the acid in the form of complexes with the titanium oxo-cluster? Why can't pristine acid be introduced into the polymer matrix? Probably, the authors want to combine the effect of obtaining ROS due to clusters, and the acid's own action? Then it is necessary to consider the removal of the solid residue from the hydrolysis of the cluster from the body.

2) The synthesis and composition of the complex:

а) Add an explanation of the origin of the water that was included in the complex.

b) The 13C-NMR spectrum (Par. 3.2, Figure S2) appears to correspond to a solution rather than a solid state, as described in the Experimental section (line 24). THF should show two signals in the spectrum, not just one.

c) The authors discuss in detail the evidence for the presence of water (lines 283-286). A 1H{13C}-NMR spectrum would be well suited for this purpose.

d) In addition, the bending vibrations of water should be in the IR spectrum. In what form was the sample used to record the IR spectra (KBr tablet?)

Further, the absence of stretching O-H vibration in the Raman spectrum (Table 2) is very strange. They must appear, and this has long been shown, for example, DOI: 10.1080/00387018708081554.

e) Give the methods and instruments used to determine the content of titanium, carbon and hydrogen. The dimension must also be specified (mass %, lines 87-88).

f) The ratio of reagents for synthesis corresponded to a twofold excess of titanium compared to the stoichiometry of the complex. Is this condition necessary for high yield? What will change if we take the stoichiometric ratio of the titanium source and Hasp?

3) Structure of the complex:

a) Powder diffraction data is needed to relate the structure determined from the single crystal to the obtained powder (1) sample.

It is likely that in the powder sample, in contrast to single crystals, there is no water. Then weak bands in the IR spectrum of the (1) powder can correspond to water sorbed by the matrix (KBr?). This would correlate with the absence of water bands in the Raman spectra.

b) Is there any influence of the "stabilizing" ligand (O2CR’) on the parameters of the core or other geometric characteristics of the complexes [Ti4O2(OR)10(O2CR’)2]?

c) Please specify which groups the atoms that form the discussed weak C85-H85E…O9[-1/2+x, 1/2-y, -1/2+z] hydrogen bond belong to.

4) Thermal Analysis

а) What is the reason for the observed large difference in thermal effects during the decomposition of a pure polymer and a composite material? It does not seem that the amount of the (1) additive is so large as to exhibit such effects. Useful information would be provided by the corresponding curves (TGA/DCS) for complex (1).

b) In Table 3, it is not clear what is "thermal transformation" and how is it different from "decomposition"?

c) What does the solid residue correspond to (calculation or characterization)?

d) DSC peaks are usually presented as onset temperatures rather than maximums.

5) Photocatalytic activity/stability of composite materials

a) The authors note that the content of the complex in the matrix does not affect the photocatalytic activity (lines 346-347). Why with an increase in the proportion of the complex, does the percentage of MB decolorization first increase and then decrease?

b) The activity results should be compared with previous authors and literature data for photocatalytic materials based on titanium/titanium dioxide/titanium compounds in a polymer matrixes.

c) The initial composite materials were characterized by the Raman method (Figure 5). Nevertheless, the materials after photocatalytic experiments were for some reason investigated by the IR method (Figure 8). What is the reason for this?

d) The discussion of the IR spectra of the materials after the experiments does not convince us of the stability of the complexes. The discussion refers to Figure 10 instead of Figure 8. In Figure 8, the absorption bands of the complex are implicit and strongly overlap with those of the polymer. Further, there is no spectrum of the polymer after exposure/treatment under the same conditions.

It would probably be more informative to involve Raman spectroscopy with mapping.

e) How comparable are the concentrations of TiO2 nanoparticles in the work 48, which give a strong band in the IR spectrum, with the expected concentration in the partial/complete decomposition of the complex in the current work?

6) Technical points:

a) Part of the Discussion is presented in continuous text, and therefore it is difficult to perceive. It is desirable to divide it into semantic paragraphs (new line) or separate subsections like the Results part.

b) One should adhere to the same principle of designating of complexes, for example [Ti4O2(OR)10(O2CR’)2], and not swap ligands in chemical formula. For example, the "new" order and notation in the formula [Ti4(μ4-O)(μ-O)(asp)2(BuiO)10] is a bit confusing. 

c) Extra signs in the inscriptions in Figure 4 should be removed.

d) Rows moved out in table 4.

e) Figure 8 does not have parts a and b, unlike the caption.

Author Response

Thank you for your in-depth review, based on which we have prepared a new version of our article. Please find attached replies to detailed remarks and comments (PDF file).

Round 2

Reviewer 4 Report

The authors responded satisfactorily to a number of comments and made some changes to the text of the manuscript. However, a number of issues still require polishing.

1) A number of previous comments on experiments/data were not reflected in the new version of the manuscript, but were only explained to the Reviewer. Adding this information will keep readers from getting confused. Please look at the previous Comments from this point of view, especially the points included in the previous Comments 2 and 3.

2) The authors agreed that their manuscript does not yet address those aspects that are reflected in the motivation for the study (Introduction, lines 56-71). It is to be hoped that further authors will obtain good results in this direction. However, the Introduction should be primarily related to the text of the current manuscript, not the future one. Thus, I insist on a reformulation/abbreviation of this "faraway" part of the Introduction (now it takes up about a third of this section). Briefly, such considerations can be presented as the future directions for the work development (Discussion/Conclusions).

3) Despite the new data (bands in IR/Raman, explanation of sample preparation), the question of the composition of the studied polycrystalline sample (1) (powder of the complex) is not clear.

In particular, X-ray diffraction analysis showed that the single crystal (1) contains water. Vibrational spectroscopy and elemental analysis have shown polycrystals (1) consistent with this.

However, NMR showed that the polycrystalline sample (1) contained THF. This is inconsistent with the above methods.

Thus, we get a contradiction in the results of identification methods. It should be resolved. If powder XRD is not available, the first steps of mass loss of (1) in TGA can be carefully considered. However, this data has not yet been provided.

I applaud the authors for honestly reflecting real results. Nevertheless, finally, it should be as clear as possible what does the polycrystalline sample includes (water only/THF only/both water and THF) and in what quantity.

4) I agree with the authors' comment about the uniqueness of the composite sample with 15% complex (1) content. In particular, it falls out of the expected trend that the solid residue in TGA should be greater, the greater the proportion of the complex (1) in the composite material (Table 3). Here the residual mass is even less than in the case of 10%-contained composite! This deviation from the logical trend must be explained. Otherwise, one has to think whether there was a confusion with the samples numbering in the study.

In addition, the fourth column in Table 3 promises to represent two quantities (Tdmax/°C / ∆m/%). However, only the temperatures are given. What about the masses?

5) Comparison of data can be done not only in review articles, but also in current studies. This helps to identify the key points that determine the investigated functional property. I can agree with the authors that it may be difficult to immediately compare their results on photocatalysis with the data of other authors due to different experiments. However, it is quite possible to compare the results with previous works of the same group of authors, where similar composite films were studied, but with a different polymer/oxo-cluster.

6) Data on the IR spectrum of a polymer that does not contain (1) but has undergone the same treatment (photocatalytic study conditions) should be provided to better understand the degradation of the material.

In addition, the addition of the IR spectrum of the material to include "trace amounts" of titanium dioxide nanoparticles would remove all questions about the clarity of evidence for the absence of (1) degradation. Otherwise, the question again arises of the amount of TiO2 nanoparticles sufficient for the manifestation of a strong vibration band, which would not overlap with the absorption of the polymer/complex (1).

7) The method for determining the titanium content should be clarified. With conventional thermogravimetry, titanium dioxide should be formed finally, not titanium. Moreover, TGA data for complex (1) were not provided. Only composite materials were examined by this method (Figure S4).

Author Response

Thank you for the valuable comments of the reviewer.
The responses to the reviewer's comments are provided below. 

1) A number of previous comments on experiments/data were not reflected in the new version of the manuscript, but were only explained to the Reviewer. Adding this information will keep readers from getting confused. Please look at the previous Comments from this point of view, especially the points included in the previous Comments 2 and 3.

It is true that answer for your comments were not reflected in the manuscript. Now, in accordance with your comments, they were taken into account in the revised version of the manuscript.

2) The authors agreed that their manuscript does not yet address those aspects that are reflected in the motivation for the study (Introduction, lines 56-71). It is to be hoped that further authors will obtain good results in this direction. However, the Introduction should be primarily related to the text of the current manuscript, not the future one. Thus, I insist on a reformulation/abbreviation of this "faraway" part of the Introduction (now it takes up about a third of this section). Briefly, such considerations can be presented as the future directions for the work development (Discussion/Conclusions).

Thank you for your valuable comments on the Introduction. This section has been revised following the reviewers' comments. In the new version, we have focused only on the issues discussed in the publication.

3) Despite the new data (bands in IR/Raman, explanation of sample preparation), the question of the composition of the studied polycrystalline sample (1) (powder of the complex) is not clear.

In particular, X-ray diffraction analysis showed that the single crystal (1) contains water. Vibrational spectroscopy and elemental analysis have shown polycrystals (1) consistent with this.

However, NMR showed that the polycrystalline sample (1) contained THF. This is inconsistent with the above methods.

Thus, we get a contradiction in the results of identification methods. It should be resolved. If powder XRD is not available, the first steps of mass loss of (1) in TGA can be carefully considered. However, this data has not yet been provided.

I applaud the authors for honestly reflecting real results. Nevertheless, finally, it should be as clear as possible what does the polycrystalline sample includes (water only/THF only/both water and THF) and in what quantity.

Sorry, but the composition of the sample is unambiguous for us and corresponds to the general formula [Ti4O2(OiBu)10(asp)2]·H2O. This was confirmed by structural data and the analysis of IR and Raman spectra. The peaks indicating the THF presence were found only in the 13C NMR spectrum of the powder sample (1). Due to the use of the 1:1 THF/HOiBu mixture as a solvent, we believe that mentioned above effect is associated with an incomplete solvent removal from the studied sample. According to the reviewer's suggestion, we re-analyzed the TG/DTG curves of (1) and the IR spectra of the volatile thermolysis products of this compound in the range of 35-150 °C (Figures S5 and S6). The thermal decomposition of (1) starts practically right after heating and is related to the detachment of H2O molecules and alcohol species (35-150 °C, Figure S5). The confirmation is the IR spectrum of volatile products of (1) thermolysis (Figure S6). The bands assigned to water vapor appear between 3400-3800 cm-1 and 1200-1900 cm-1. While, bands, which appeared at ~3450 cm-1, ~2990, 2800 cm-1, and 1030 cm-1, can be attributed to stretching vibrations of -OH, -CH, and -C-O groups, respectively, and indicates the detachment of alcohol species. If the THF would appear in the gas phase in the range of 35-150 °C, in the IR spectrum should appear the bands at ~ 2970, 2860 cm-1, ~ 1100 cm-1, and ~900 cm-1 assigned to v(CH), v(C-O-C), and v(C-C), respectively.

4) I agree with the authors' comment about the uniqueness of the composite sample with 15% complex (1) content. In particular, it falls out of the expected trend that the solid residue in TGA should be greater, the greater the proportion of the complex (1) in the composite material (Table 3). Here the residual mass is even less than in the case of 10%-contained composite! This deviation from the logical trend must be explained. Otherwise, one has to think whether there was a confusion with the samples numbering in the study.

You're absolutely right, TGA results for PCL + 10 wt.% and PCL + 15 wt.% samples were mixed up (sample markings have been misread). Thank you for paying attention.

In addition, the fourth column in Table 3 promises to represent two quantities (Tdmax/°C / ∆m/%). However, only the temperatures are given. What about the masses?

This error has been corrected.

 5) Comparison of data can be done not only in review articles, but also in current studies. This helps to identify the key points that determine the investigated functional property. I can agree with the authors that it may be difficult to immediately compare their results on photocatalysis with the data of other authors due to different experiments. However, it is quite possible to compare the results with previous works of the same group of authors, where similar composite films were studied, but with a different polymer/oxo-cluster.

The results of research on the photocatalytic activity of the PCL + (1) composite may be compared with the results of our previous investigations concerning PCL + TOCs composites (TOCs = [Ti4O2(OiBu)10(O2CR’)2] (R’ = PhNH2 and C13H9)) [8]. However, in our previous tests, the composite fittings (produced by the injection molding method ) were used, and the photocatalytic process was conducted for 30 hours in visible light. The comparison of the obtained results shows that after 30 h, the photocatalytic activity of PCL + (1) is comparable to the PCL + ([Ti4O2(OiBu)10(O2CNH2)2]) system and weaker than for PCL+ ([Ti4O2(OiBu)10(O2CC13H9)2]). However, if we compare the test results of PCL + (1) film samples with PMMA + TOCs films (TOCs = [Ti4O2(OiBu)10(O2CR’)2] (R’ = PhNH2 and C13H9), PMMA = poly(methyl methacrylate)), no significant differences were found.

This fragment was added to the Discussion part of the manuscript

6) Data on the IR spectrum of a polymer that does not contain (1) but has undergone the same treatment (photocatalytic study conditions) should be provided to better understand the degradation of the material.

In addition, the addition of the IR spectrum of the material to include "trace amounts" of titanium dioxide nanoparticles would remove all questions about the clarity of evidence for the absence of (1) degradation. Otherwise, the question again arises of the amount of TiO2 nanoparticles sufficient for the manifestation of a strong vibration band, which would not overlap with the absorption of the polymer/complex (1).

Considering your remarks, IR spectra of a PCL before and after 140 h of the photocatalytic process is presented in Figure S8.

Analysis of IR spectra confirms the effect of degradation which was earlier found in PCL + (1) composite spectra. This is evidenced by: (a) the increase in the band attributed to the stretching vibrations of -OH (~3320 cm-1) groups, (b) the strong band of v(COO) at ~1700 cm-1, (c) the appearance of the -OH bending modes at ~1410 cm-1, (d) the width increase of bans between 1000-1150 cm-1 assigned to COC modes of ester groups. The difference between IR spectra of PCL and PCL + (1) concerning the bands coming from water molecules, i.e. 3100-3700 cm-1. One of the reasons may be the difference in the drying method of both samples after the photocatalytic processes. However, this effect will be explained in detail during the following stages of our research, during which the degradation process of PCL + (1) composites will be studied.

Figure S7 shows the IR spectrum of the PCL + 0.5 TiO2 wt.% system, which was compared with the spectra of the complex (1), PCL, and the composite PCL + (1) 15 wt.%. Two bands appearing in the PCL + TiO2 composite spectrum at c.a. 720 and 560 cm-1 were assigned to stretching vibrations of titanium dioxide (according to the literature data, the positions of these bands are similar to those found in spectra of TiO2 nanoparticles [54, 55]). These bands are superimposed on the PCL polymer bands and their degradation products and the complex (1) appearing in this range. Considering these data, we can state that the hydrolysis result of (1) should be the appearance of a clear band at 720 cm-1, which in our spectra of PCL + (1) composites after photocatalytical process (140 h) was not found (Figure 8).

Figure S8 and a short commentary are available in the Discussion of the manuscript revised version.

7) The method for determining the titanium content should be clarified. With conventional thermogravimetry, titanium dioxide should be formed finally, not titanium. Moreover, TGA data for complex (1) were not provided. Only composite materials were examined by this method (Figure S4).

We apologize for the mistake, we have made. Titanium content has been gravimetrically determined as TiO2, according to the Meth-Cohn et al. method [27; Meth-Cohn, O., Thorpe , D.; H. J. Twitchett, H.J. Insertion reactions of titanium alkoxides with isocyanates and carbodiimides J. Chem. Soc. C, 1970, 132-135; Doi: 10.1039/J39700000132.]. We have made appropriate changes to the new version of the manuscript.